# Use of Personalized Biomarkers in Metastatic Colorectal Cancer and the Impact of AI

**DOI:** 10.3390/cancers14194834

**Published:** 2022-10-03

**Authors:** Simona-Ruxandra Volovat, Iolanda Augustin, Daniela Zob, Diana Boboc, Florin Amurariti, Constantin Volovat, Cipriana Stefanescu, Cati Raluca Stolniceanu, Manuela Ciocoiu, Eduard Alexandru Dumitras, Mihai Danciu, Delia Gabriela Ciobanu Apostol, Vasile Drug, Sinziana Al Shurbaji, Lucia-Georgiana Coca, Florin Leon, Adrian Iftene, Paul-Corneliu Herghelegiu

**Affiliations:** 1Department of Medical Oncology-Radiotherapy, “Grigore T. Popa” University of Medicine and Pharmacy, 16 University Str., 700115 Iasi, Romania; 2Department of Medical Oncology, AI.Trestioreanu Institute of Oncology, 022328 Bucharest, Romania; 3Department of Medical Oncology, “Euroclinic” Center of Oncology, 2 Vasile Conta Str., 700106 Iasi, Romania; 4Department of Biophysics and Medical Physics-Nuclear Medicine, “Grigore T. Popa” University of Medicine and Pharmacy, 16 University Str., 700115 Iasi, Romania; 5Department of Pathophysiology, “Grigore T. Popa” University of Medicine and Pharmacy, 700115 Iasi, Romania; 6Department of Anesthesiology and Intensive Care, Regional Institute of Oncology, 700115 Iasi, Romania; 7Pathology Department, “Grigore T. Popa” University of Medicine and Pharmacy, 700115 Iasi, Romania; 8Department of Gastroenterology, “Grigore T. Popa” University of Medicine and Pharmacy, 16 University Str., 700115 Iasi, Romania; 9Gastroenterology Clinic, Institute of Gastroenterology and Hepatology, ‘St. Spiridon’ Clinical Hospital, 700115 Iasi, Romania; 10Faculty of Computer Science, Alexandru Ioan Cuza University, 700115 Iasi, Romania; 11Faculty of Automatic Control and Computer Engineering, Gheorghe Asachi Technical University, 700115 Iasi, Romania

**Keywords:** metastatic colorectal cancer, ncRNA, predictive biomarkers, prognostic biomarkers, artificial intelligence

## Abstract

**Simple Summary:**

Colorectal cancer is one of the most frequent cancers worldwide, with a high incidence and mortality. Although many treatment options are available for metastatic disease, patient survival is still limited. The molecular classification of colorectal cancer proposed in 2015 has helped us to better understand colorectal cancer and realize a more effective implementation of therapeutic sequences. It has also been observed that the existing mutational landscape is closely correlated with the epigenetics of colorectal cancer. The identification of prognostic and predictive biomarkers in this context becomes a necessity closely related to therapeutics, and artificial intelligence can be used to discover new biomarkers.

**Abstract:**

Colorectal cancer is a major cause of cancer-related death worldwide and is correlated with genetic and epigenetic alterations in the colonic epithelium. Genetic changes play a major role in the pathophysiology of colorectal cancer through the development of gene mutations, but recent research has shown an important role for epigenetic alterations. In this review, we try to describe the current knowledge about epigenetic alterations, including DNA methylation and histone modifications, as well as the role of non-coding RNAs as epigenetic regulators and the prognostic and predictive biomarkers in metastatic colorectal disease that can allow increases in the effectiveness of treatments. Additionally, the intestinal microbiota’s composition can be an important biomarker for the response to strategies based on the immunotherapy of CRC. The identification of biomarkers in mCRC can be enhanced by developing artificial intelligence programs. We present the actual models that implement AI technology as a bridge connecting ncRNAs with tumors and conducted some experiments to improve the quality of the model used as well as the speed of the model that provides answers to users. In order to carry out this task, we implemented six algorithms: the naive Bayes classifier, the random forest classifier, the decision tree classifier, gradient boosted trees, logistic regression and SVM.

## 1. Introduction

Colorectal cancer (CRC) is the third most frequent cancer worldwide in both sexes, with mortality rates of 45%, 35% and 47.8% in Europe, the USA and worldwide, respectively [1]. However, CRC is a heterogeneous disease with widely variable clinical outcomes, in terms of both prognosis and drug response. This is the reason for developing effective treatments for patients with CRC, i.e., to prolong survival in metastatic settings. In order to make treatments more efficient, it is very important to identify the prognostic and predictive markers, to allow the efficient targeting of the tumor cells. Epigenetics, defined as alterations in gene expression, play a central role in the pathogenesis of various cancers, including CRC. In fact, there are several markers used to monitor metastatic colon cancer, but studies in recent decades have shown promising possibilities for using epigenetic biomarkers, given the interaction of ncRNA with different gene mutations involved in CRC pathogenesis.

## 2. Genomics in mCRC

Various genomic alterations have been studied in colorectal cancer, as the development of both predictive and prognostic biomarkers is important in personalized medicine and can be incorporated in treatment decisions.

Mismatch repair deficiency and microsatellite instability (MSI) are frequently associated with Lynch syndrome, in up to 20% of colorectal cancers, and are defined by mutations in mismatch repair (MMR) genes [2], making the cell unable to correct DNA errors. MMR deficiency is characterized by germline or somatic DNA alterations in MMR genes (MLH1, MSH2, MSH6 or PMS2), leading to colorectal cancer. Most frequently, the loss of MLH1 expression leads to sporadic colorectal cancer [3]. MSI status is evaluated in early-stage colorectal cancer and is a predictive biomarker for immunotherapy with pembrolizumab in stage IV disease [4]. Moreover, the MSI status can provide prognostic information, as patients with tumors that are dMMR (MSI-high) show longer survival [5], as well as patients with proximal tumors associated with MSI [6]. In metastatic setting, MSI-H tumors appear to behave more aggressively and have a negative impact on survival [7].

BRAF V600E mutations are downstream targets of the RAS signaling pathway and are altered in 10% of colorectal cancer (CRC) patients; these mutations are also mutually exclusive with the KRAS mutation [8]. Patients with these mutations have lower survival rates when they are associated with MSI-low tumors, and current research approaches include combining BRAF inhibitors with agents that block other signaling pathways. Although BRAF inhibitors are effective treatments in BRAF mutant melanoma, this approach has been ineffective in colorectal cancer. Preclinical studies suggest that could be due to a rapid ERK reactivation [9]. Available data suggest that BRAF mutations are associated with resistance to anti-EGFR therapy [10]. Combinations with MEK inhibitors have demonstrated improved PFS and ORR compared to cetuximab and chemotherapy and could be an option for later lines of treatment [11].

KRAS and NRAS mutations are the most prevalent forms of genomic alteration; they are found in 75% of CRCs and are associated with a worse prognosis and resistance to anti-EGFR therapy [12]. Studies have shown that the presence of KRAS mutations lead to a worse survival when anti-EGFR therapy such as cetuximab or panitumumab are added to the chemotherapy regimen in metastatic setting [13,14]. 

A new biomarker is represented by KRASG12C mutation, found in 14% of non-small cell lung cancer (NSCLC) and 3% of CRC. Two new molecules, sotorasib and adagrasib were found to decrese the phosphorylation of ERK and promote the tumor regression in mice bearing KRAS G12C-mutant NSCLC tumors [15,16].

In a phase 1 study, sotorasib was evaluated in patients with refractory KRAS G12C-mutated solid tumors (NCT 03600883). In mCRC cohort, the objective response rate (ORR) was 7.1% and the disease control rate (DCR) was 73.8%. The median PFS in this group was 4 months [17]. In the phase 2 CodeBreak 100 (NCT03600883) trial studied sotorasib in patients with metastatic KRASG12C-mutant CRC who had progressed on prior chemotherapy treatment and the ORR was 9.7% and the DCR was 82.3% [18].

The KRYSTAL-1 study (NCT03785249) is a phase 1/2 study investigating adagrasib monotherapy in patients with advanced or metastatic solid tumors harboring a KRAS G12C mutation and previously treated with chemotherapy and/or anti PD-L1 therapy. In the CRC cohort, the disease control rate was 87% and progression-free survival was 5.6 months. One of two patients achieved a partial response (duration of response, 4.2 months) [19].

Some cohorts in the CodeBreak 101 umbrella trial (NCT04185883) combine sotorasib with other approved agents including a PD1/PD-L1 inhibitors, an mTOR inhibitor, MEK inhibitor, a CDK 4/6 inhibitor, a VEGF inhibitor with various chemotherapies. The KRYSTAL-1 umbrella trial is also including similar strategies. However, adding Palbociclib to KRAS G12C inhibitors in preclinical studies, showed significantly more suppression of RAS pathway phosphorylation, cell-division genes, and cell-cycle progression [20].

In combinations with immunotherapy, targeted therapy or KRAS-G12C inhibitors failed to provide significant clinical benefit due to the complexity of the signaling pathway [21]. 

HER2 alterations occur in 2–6% of metastatic CRCs and confer resistance to treatment with EGFR inhibitors [22]. Efficient treatment options targeting HER2 in other tumors such as gastric cancer or breast cancer supports the role of HER2 as a predictive biomarker. Anti-HER2-targeted therapy has been proven effective in this setting; for example, response rates of up to 38% for the use of trastuzumab, TDM-1 and pan-HER2 inhibitors such as neratinib or lapatinib [23,24] have been documented.

NTRK fusions involve three genes that encode transmembrane receptors. NTRK inhibitors such as entrectinib and larotrectinib have been associated with tumor responses in CRC patients [25].

PI3K mutations have been described in KRAS-wild-type CRC and are responsive to anti-EGFR therapy [26]. Moreover, they are associated with a negative prognosis in BRAF-wild-type tumors [27], especially those showing mutations in exons 9 and 20.

Several genomic alterations have been evaluated as predictive biomarkers for the response to chemotherapy, such as those involving dihydropyrimidine dehydrogenase (DPD), thymidylate synthetase (TS) expression and UDP-glucuronosyltransferase 1A1 (UGT1A1). DPD deficiency has been associated with increased fluoropyrimidine toxicity; thus, it has potential predictive value in clinical settings. It causes a deficit in the metabolism of thymine and uracil, resulting in accumulation in the blood and resulting in increased toxicity. Current guidelines are conflicting in recommending DPD genotyping before fluoropyrimidine-based therapy [28]. Currently, data on its prognostic value are limited [29]. UGT1A1 expression has been associated with increased SN-38, leading to increased toxicity in irinotecan-based chemotherapy. Irinotecan is metabolized into the active form, SN-38, leading to severe treatment hematologic and digestive toxicity [30]. However, UGT1A1 genotyping is not routinely applied in clinical settings [31]. 

TS and ERCC1 expression levels have been described as potential biomarkers in CRC. ERCC1 is involved in the cellular response to DNA damage, and TS has been shown to be predictive of responses to fluoropyrimidine chemotherapy. Low TS levels are associated with improved response rates and OS in patients treated with a FOLFOX regimen [32].

Molecular profiling using liquid biopsies has been validated in various tumor types in clinical settings and can be used to assess circulating tumor cells (CTCs), circulating tumor DNA (ctDNA) and exosomes released by cancer cells. However, its implementation in clinical practice remains technically challenging. Several studies have shown ctDNA to have both prognostic and predictive value in clinical settings [33]. A reduction in ctDNA levels of at least 80% has been associated with a favorable response rate, and variations in ctDNA after the initial treatment response could predict clinical relapse within several months [34]. 

The main advantage of the liquid biopsy is that ctDNA captures alterations of many genes, specifically EGFR, ERBB2, PIK3CA or MAP2K1, revealing new potential targets for therapies such as anti-BRAF, anti-EGFR and anti-HER2 agents. In metastatic CRC, ctDNA can represent an important tool to monitor the molecular evolution of CRC over time, during the different courses of treatment. Quantitative and qualitative fluctuation of molecular landscapes, revealed by ctDNA, suggesting a molecular evolution of CRC, which would have been difficult to assess by tissue biopsy were found [35,36,37].

The pulsatile behavior of tumor-specific mutant clones, detected through mutation monitoring over time on ctDNA, provided a scientific rational for the retreatment with anti-EGFR. In CHRONOS trial (NCT03227926), the mCRC patients approaching third or later line of treatment were assessed in ctDNA for RAS, BRAF and EGFR ectodomain status and re-challenged with anti-EGFR therapy (panitumumab) only for the patients with a mutation-negative status. A 30% response rate and a 63% disease control rate was reported, demonstrating that genotyping tumor DNA in the blood of CRC patients can be used to direct therapy and can be included in the management of advanced CRC patients [38,39].

Tumor mutational burden (TMB) in CRC is typically increased in case of microsatellite instability (MSI) or pathogenic mutations occurring in domains of the DNA polymerases POLE and POLD, being correlated with the response to immunotherapy. Recently, Food and Drug Administration (FDA) approved TMB as a companion biomarker for the treatment with pembrolizumab or dostarlimab in mCRC [40].

The golden standard for TMB evaluation is represented by tumor-tissue specimens [41], but the intratumoral heterogeneity represents a limit for TMB evaluation, supporting the role of ctDNA as a monitoring biomarker, being known that TMB can change under treatment with standard cytotoxic agents in CRC [42].

In the ARETHUSA trial (NCT03519412) the metastatic-colorectal patients who failed standard therapies undergo treatment with pembrolizumab, are tested for o6-methylguanine-DNA-methyltransferase (MGMT) expression (IHC), then for MGMT promoter methylation [43].

The microsatellite instability (MSI) also represents a relevant biomarker for immunotherapy sensitivity in CRC, but similarly to TMB, MSI status is subjected to both spatial and temporal heterogeneity, making its monitoring through ctDNA therapeutically valuable [44].

## 3. Transcriptomics in mCRC: Immunoscore

The classification of colorectal cancer plays an essential role in establishing the prognosis and the choice of therapeutic management for the patient. The TNM classification is the system most commonly used to determine the progression of CRC, but a more in-depth approach is needed to establish the prognosis and therapeutic strategy.

In 2015, the International Consortium of CRC Subtypes proposed a unified transcriptomic classification that allowed the identification of four biologically distinct consensus molecular subtypes (CMS), which subsequently allowed the classification of CRC into four subtypes with distinct molecular and biological characteristics: CMS1 (immune to microsatellite instability), CMS2 (canonical), CMS3 (metabolic) and CMS4 (mesenchymal) [45].

### 3.1. Clinical and Prognostic Associations of the Consensus Molecular Subtypes

#### 3.1.1. CMS1

Serrated polyps, precursor lesions of CMS 1 have an evolution to carcinoma characterized by their high mutation rate for BRAF V600E, hypermethylation of CpG islands with the loss of the tumor suppressor function, a defective DNA mismatch repair (MMR) system and tumor microenvironment with lymphocytic infiltration. The hypermethylation of or mutations in the MMR gene promoter regions can cause microsatellite instability (MSI). MSI cancers have approximately 47 mutations per 10^6^ bases, compared to stable microsatellite tumors (MSS or CMS2), which have an average of 2.8 per 10⁶ bases [46,47]. 

Clinical Implications—A favorable prognosis can be given when there is a presence of specific populations of T cells: cytotoxic CD8+ T lymphocytes, CD4+ activated Th1 type helper T cells (Th1) and natural killer cells. CMS1 tumors are associated with poorer survival after recurrence [45,48,49,50,51].

#### 3.1.2. CMS2

CRC in the CMS2 category belongs to the canonical adenoma–carcinoma sequence [52]. The gene expression profile is directly related to a differentiated epithelial cell phenotype characterized by the loss of the APC tumor suppressor gene, and an activating mutation in KRAS. This may produce high rates of chromosomal instability (CIN), because of the loss and/or gain of large portions of chromosomes, the loss of heterozygosity and aneuploidy [53]. In the case of CMS2 tumors, the Wnt–β-catenin and MYC signaling pathways are active. Recently, it has been suggested that the precursor lesions of the mutant CRC KRAS are tubular hairy adenomas with serrated features and mixed histological variants between CMS1 and 2 [54].

Clinical Implications—The five-year overall survival for all the stages of CMS2 is the highest of any subtype [45]. In the case of CMS2 cancers, lesions on the left side are found more frequently (59%), which leads to higher survival rates after recurrence (35 months). CMS1 tumors commonly occur in the right colon and lead to poor survival after recurrence (9 months) [45].

#### 3.1.3. CMS3

Of all the subtypes, CMS3 is most similar to normal colonic tissue in terms of gene expression, but pathway analysis revealed that CMS3 RNA is enriched for the majority of the metabolic pathways investigated, including glutamine, fatty acids and lysophosphatidic metabolism. KRAS mutants are present in each molecular subtype, but they are more prevalent among CMS3 CRC (68%) [54]. 

Clinical Implications—In the case of metastatic CRC, the higher frequency of KRAS mutations among these tumors reduces standard chemotherapeutic options, as mutant KRAS is usually an indicator of poor response to epidermal growth factor receptor (EGFR) monoclonal antibodies (mAbs, e.g., cetuximab) [55,56]. In the case of CMS3 tumors that do not show KRAS mutations (neither BRAF nor PIK3CA), EGFR mAbs may be useful.

#### 3.1.4. CMS4

Experimental studies showed that premalignant human organoid cultures with the genetic background of a serrated adenoma (BRAF V600E) exposed to a high or low level of transforming growth factor β (TGF-β) in the microenvironment, developed into a CMS4 (mesenchymal) or CMS1 (MSI) phenotype in response to, respectively [57]. The CMS4 tumor microenvironment is proinflammatory, with the presence of Treg cells, T-helper 17 cells, myeloid-derived suppressor cells, and tumor-promoting macrophages [58]. The presence of immunosuppressive cytokines such as IL-23 and IL-17 links CMS4 cancers to colitis-associated colorectal carcinoma, where TP53 inactivation occurs early in the transformation to dysplasia [49], which is distinct from CMS2 precursor lesions, where the loss of the TP53 tumor suppressor function occurs late in the adenoma-to-carcinoma sequence. CMS4 tumors exhibit extremely low levels of hypermutation, an MSS status, and very high SCNA counts.

Clinical Implications—CMS4 cancers are often diagnosed in advanced stages, leading to a poor prognosis, with the worst overall survival at 5 years (62%). For metastatic disease, CMS4 cancers are resistant to anti-EGFR therapy, regardless of the state of the KRAS mutation [58,59,60]. Antiangiogenic therapies, such as bevacizumab, are standard supplements for stage IV disease [61]; however, other stromal elements, such as CAFs and pro-tumorigenic immune cells, such as tumor-associated macrophages, are not specifically targeted.

### 3.2. Immunoscore (IS)

Present (as well as past) official cancer-classification approaches (American Joint Committee on Cancer/Union for International Cancer Control, AJCC/UICC) still take into consideration only the tumor characteristics, (TNM staging) [62]. However, immune system cells seem to play a special role through the components of the tumor microenvironment (TME) and can interfere with the personalized evolution of the neoplasia [63], which is a good explanation for studies that showed how, among patients within the same calculated TNM stage, the clinical outcome could be very different [64,65,66]. It appears that, in most solid tumors, high T-cell infiltration is associated with a decreased risk of tumor dissemination and improved survival. Until now, this correlation had been documented in CC, but also in melanoma and ovarian, breast, prostate and lung cancers [67].

IS was proposed as a standardized diagnostic, tissue and digital-pathology-based test, and was considered a system for assessing tumor prognosis and the risk of recurrence in relation to the patient’s particular immune context. It provides a personalized score defined by the precise quantification (assessing the densities) and the identification of two lymphocyte populations, the CD3^+^ and CD8^+^ T cells, in the tumor core and its invasive margin (Figure 1). The densities of CD3CT, CD3IM, CD8CT, and CD8IM are reported and converted into percentile values, determined by international validation studies [68,69,70]. The mean percentile for the four markers is calculated to generate the IS percentile value and translated into a three-category scoring system: IS high, IS intermediate and IS low [67].

As studies demonstrated, IS could constitute the first highly efficient immune-based scoring system for cancer and a personalized biomarker with prognostic value superior to that of the TNM staging system [71].

There are currently AI studies that propose the correlation of IS with other parameters of the patient, for the purpose of diagnosis and a personalized therapeutic approach [72]. It seems that, using artificial intelligence tools, additional prognostic markers (to IS) could be detected on pathological slides, the most promising of which is CD3 (using a single standard CD3 pathological slide) [73]. Studies showed that both tumor stroma and tumor cell intrinsic variables, in association with immune cell infiltrates, should be taken into account during colorectal cancer prognosis.

In conclusion, the evaluation of cancer patients using a score that takes into account both the molecular histopathologic subtype (as TNM does) and the specific immune context, could be the basis for a complete future personalized approach in the evolution of CRC, as well as for other cancer types. In the context of the complexity of the elements of the tumor–immune interaction, AI demonstrates, once more for this topic, that it can help to improve patient care by assisting the practitioner using a personalized approach to decision making related to CRC patients’ diagnosis, prognosis and treatment.

## 4. Epigenomics in mCRC

It is well established that a significant part of the pathogenesis of cancer, including colorectal cancer, can be explained by epigenetic modifications, such as DNA methylation and histone modifications, and epigenetic regulators, including ncRNAs.

### 4.1. Histone Modifications

In non-dividing cells, the DNA is wrapped around nucleosomes, an octameric protein structure comprising four pairs of core histones. Each histone core has an individual tail that contains lysine and arginine residues. The tail is subject to posttranslational modifications, which can influence gene expression. Several types of histone modifications are involved in crucial cellular functions, such as growth and differentiation. Histone modifications are produced through the mutation of catalyzer enzymes that intervene in the post-translation phase. The most frequently affected are histone deacetylases (HDACs) and histone acetyltransferases (HATs). 

Reversible histone acetylation is an active process that is achieved by the addition or removal of histone acetyltransferases (HATs) and deacetylases (HDACs). Recently, the HATs identified mainly include P300/CBP, GNAT, MYST, P160, PCAF, and TAFII230 families. HDACs can be classified into four groups on the basis of their homology with the original yeast enzyme sequence. Among these HDACs Class I, II and IV are zinc-dependent, while Class III are NAD+ -dependent [74]. HATs transfer the acetyl group of acetyl coenzyme A to the terminal of histone amino acid and relax the structure of chromatin under the action of electric charge, which is helpful to transcription employing increased accessibility of DNA. On the contrary, HDAC removes the terminal acetyl group of histone lysine, making the structure of chromatin is compact, which results in the inhibition of transcription. In general, hyperacetylation leads to increased gene expression, which is related to the activation of gene transcription, while hypoacetylation means repression of gene expression [75].

Since research on the role of histone modifications in the development of colorectal cancer (CRC) began, the apparent relationship between the former and latter has changed. It has been found that changes in histone modification patterns can impair the expression of genes that play crucial roles in the development of colorectal cancer. Karczmarski et al. showed that H3K27 acetylation was significantly increased in CRC samples compared to normal tissue [76]. In addition, several reports have indicated that global histone acetylation has been positively correlated with the tumor stage, lymphatic metastases, poor survival, unfavorable prognoses, histological subtypes and cancer recurrence [77]. Through multivariate analysis, Hashimoto et al. found that the global upregulation of acetylated histone H3 (H3Ac) expression in colorectal cancer tissues was related to poor overall survival [78]. Benard et al. showed that the increased nuclear expression of H3K56ac and H4K16ac correlated with higher survival rates for CRC patients and a lower chance of tumor recurrence [79]. Ashktorab et al. demonstrated that the acetylation of H3K12ac and H3K18ac was significantly increased in moderate to well-differentiated colon cancer and decreased in poorly differentiated colon cancer. They also observed the presence of high levels of HDAC2 in adenocarcinoma compared to those in adenoma, suggesting that HDAC2 expression is closely related to the progression from adenoma to adenocarcinoma [80].

In addition, there are some studies that have indicated that certain histone acetylations may be targeted by a specific signaling pathway [77]. For example, Liu et al. found that RAS–PI3K signaling downregulated the level of H3K56ac, which is linked to the transcription, proliferation and migration of cancer cells [81]. A study by Zhang et al. found that cell-cycle-related and growth-enhancing protein (CREPT) cooperated with P300 acetyltransferase and stimulated Wnt/β-catenin signaling to promote H4Ac and H3K27ac expression [82]. Tamagawa et al. showed that the methylation of H3K27me2 in liver metastasis was decreased compared to that in the primary tumor, whereas the expression of H3K36me2 showed the opposite pattern. They also demonstrated a positive correlation between the expression of H3K37me2 and tumor size, with lower survival rates. The expression of H3K37me2 could be used as an independent prognostic factor for CRC patients with metachronous liver metastasis [83]. Kornbluht et al. described how lower levels of the histone methyltransferase SEDT2 facilitated CRC development by affecting alternative splicing [84]. Another study conducted by Qin showed that the expression of G9A was increased in CRC tumor tissues, and the overexpression of G9A was mainly correlated with stage, tumor differentiation, and tumor relapse in CRC [85]. The elevated expression of lysine-specific demethylase (LSD1) observed in colon cancer tissues was strongly correlated with advanced TNM stages and distant metastasis [86]. To date, there has been little research on the relationship between histone phosphorylation and colorectal cancer. Several studies have shown that histone phosphorylation aberrations are correlated with the pathogenesis of colorectal cancer [77]. For example, a decrease in double-specific phosphatase 22 expression (DUSP22) was observed in colorectal cancer specimens, and the reduced expression of DUSP22 in stage IV patients was linked to poor survival rates [87]. Lee et al. revealed that the phosphorylation of histone H2AX (p-H2AX) was found to be increased in CRC tissues and was correlated with more aggressive tumor behavior as well as poor survival for CRC patients [88]. A recent investigation revealed that the process of histone modification is reversible, and aberrations can be restored to an almost normal status through epigenetic therapy. Thus, histone modification serves as a promising therapeutic target in treating various cancers in combination with conventional treatment. The current investigation indicated that the deregulation of HATs and histone HDACs was involved in the progression of a range of cancers, making them attract considerable interest from the research community. Thus, various HDACis and histone deacetylase inhibitors (HDIs) have become favored in attempts to attenuate many human cancers, including colorectal cancer [77]. 

The research undertaken concluded that the process of histone modification is reversible and their aberrant modification can be reestablished to a nearly normal status through epigenetic therapy. Therefore, histone modification represents a promising therapeutic target in the treatment of various cancers. Histone deacetylation and methylation inhibitors are the most promising in colorectal cancer. 

Aberrant histone methylation is a frequent result of gene mutation and is associated with the occurrence and development of cancer. The development of new cancer treatments includes studies that are trying to identify molecules targeting histone methyltransferases or demethylases, therefore most of the histone modifying enzymes serve as a drug target [75]. 

Until now, four HDACis have been approved by Food and Drug Administration (FDA) for the treatment of patients with cutaneous T cell lymphoma and peripheral T cell lymphoma [89,90]. The HDACis applied to colorectal cancer is expanding fast, with a wide list of candidates that are ongoing study and clinical trials.

### DNA Methylation

DNA methylation is a frequently used signaling tool that can “switch off” gene expression. The process refers to the conversion of the cytosine ring to 5-methylcytosine by the addition of a methyl group to the DNA strand. The reaction is catalyzed by DNA methyltransferases (DNMTs) and is reversible. The 5-methylcytosine residues are usually positioned next to a guanine base (CpG methylation), resulting in a structure that blocks both DNA strands. DNA methylation is present across the genome, with the exception of certain areas where the content of CpG is high (promoter areas called CpG islands). The methylation of CpG islands (called hypermethylation) can result in the inappropriate silencing of genes such as tumor suppressor genes. Compared to that of normal cells, the genome of malignant cells appears to be hypomethylated overall, but with the hypermethylation of genes controlling the cell cycle, invasion, DNA repair, and other processes where silencing would lead to the promotion of cancer. In colon cancer, hypermethylation can be detected in the early stages [91]. This evidence is solid in the case of CRC, in which aberrant hypermethylation has been identified in the promoter regions of essential tumor suppressor genes [92], including CDKN2A (in the promoters p16INK4A and p14ARF) [93,94], MLH1 [95] and APC [96].

Hypermethylation outside the CpG islands, especially in gene bodies, appears to be positively correlated with gene expression. Genome hypomethylation was one of the first aberrant methylation events reported in CRC and is an early event in colorectal carcinogenesis. Indeed, hypomethylation has been observed in various stages of the disease, from early adenomas to adenocarcinomas and metastases, with a linear correlation between the degree of demethylation and the stage of the disease [97]. Because the overall loss of DNA methylation has also been described during normal aging and senescence, its role in carcinogenesis (and, therefore, as an independent risk factor) is the subject of an ongoing debate. However, DNA methylation has been thought to be the missing link that explains why cancer is an age-related disease [98]. In general, the hypomethylation of DNA at three specific sites has been linked to proto-oncogene activation in CRC, including in the promoter regions, which may lead to the loss of gene imprinting (e.g., IGF2) [99] or the direct activation of proto-oncogenes (for example, MYC and HRAS) [100] and distant regulatory regions, such as super intensifiers and antisense promoters located downstream in certain repetitive elements (such as long intercalated element 1 (LINE-1)), whose expression is reduced to evolutionary silence under normal physiological conditions [101]. Because up to 17% of the human genome consists of LINE-1 elements, their hypomethylation has been used as a surrogate for global DNA hypomethylation and is associated with early-onset CRC and poor prognosis, making LINE-1 a potential biomarker [102]. These LINE-1 elements, if activated by hypomethylation, can also function as retrotransposons through a “cut and paste” mechanism, inserting themselves into distant fragile places (unstable genomic regions) and leading to genomic instability. Consequently, LINE-1 hypomethylation is inversely correlated with MSI and the CpG island methylator phenotype (CIMP) [103].

Nowadays, it is accepted that, beside genetic mutations, epigenetic mechanisms, such as aberrant DNA methylation, are involved in every step of cancer development and progression. The old methods cannot predict the prognosis of particular cases, but for clinicians it is important to be able to accurately know which patient is at high risk for recurrence and benefit from chemotherapy and to which chemotherapy. In consequence, it is essential to find novel biomarkers that would help clinicians in the decision-making process in the management of patients. Different technologies, such as methylation microarrays and next generation sequencing, helped in the advancement of our understanding of epigenetic events. Epigenetic signatures, for example, neoplasm-specific panels of methylated genes or specific miRNAs profiles, represent the future in the early diagnosis and prognosis prediction of CRC patients [36,104].

### 4.2. miRNA

MiRNAs are small non-coding RNAs that originate in larger transcripts and have a role in messenger RNA (mRNA) regulation in the cytoplasm [105,106]. MiRNAs have multi-target potential; for instance, they are able to target a single miRNA by inducing translational repression. Additionally, miRNAs can undergo mRNA cleavage and consequent decay, which can target up to 200 mRNAs or hundreds of target genes, followed by a lower expression of the protein levels, while different miRNAs can modulate the same mRNA target [107]. They are highly stable molecules, with relatively high specificity in cells and tissues, and are easily determined in various biological samples from tissues to saliva, serum, circulating exosomes and feces. They have emerged as good non-invasive biomarker candidates for the monitoring of metastatic disease or the prediction of the response in colorectal cancer [106]. MiRNAs are involved in different molecular pathways of the carcinogenesis of colorectal cancer such as the Wnt/β-catenin, TGF-β and EGFR pathways or epithelial-to-mesenchymal transition (EMT), and are correlated with the evolution of metastatic disease or with the efficacy of systemic treatments. 

WNT/β-catenin pathway

The dysregulation of the Wnt/β-catenin pathway is involved in CRC carcinogenesis from an early stage, through the upregulation of the expression of Wnt target genes via β-catenin [108].

The crosstalk between miRNAs and the Wnt/β-catenin pathway was demonstrated, on the one hand, by miRNAs that activate/inhibit the canonical Wnt pathway and, on the other hand, by the activation of the Wnt pathway, which increases the expression of miRNAs. The regulatory effect of miR-224 on GSK3β and SFRP2 genes is followed by the activation of Wnt/β-catenin signaling and the nuclear translocation of β-catenin [109].

The suppression of miR-224 can restore the expression of SFRP2 and GSK3β and suppress Wnt/β-catenin-mediated cell metastasis and cell proliferation. The Wnt pathway can also stabilize β-catenin in colorectal cancer, thereby modulating the symmetrical cell division of cancer stem cells (CSCs), enhancing the initiation and progression of CRC. Snail is an epithelial–mesenchymal transition (EMT) inducer that regulates the symmetrical cell division of cancer stem cells (CSCs) and enhances the expression of microRNA-146a (miR-146a) through the β-catenin–TCF4 complex. MiR-146a maintains Wnt activity and directs symmetrical division by targeting NUMB to stabilize β-catenin. The miR-146a–NUMB axis was demonstrated as being able to regulate the Wnt pathway in colorectal cancer stem cells (CRCSCs), rather than the Notch pathway, and inhibiting Wnt activity has a similar effect on MEK inhibition. Interference with the Snail–miR-146a–β-catenin loop by inhibiting MEK or Wnt activity reduces the symmetrical division of CRCSCs, attenuating the tumorigenicity, and the high-Snail–low-NUMB profile has been correlated with cetuximab resistance and a poorer prognosis in colorectal cancer [110].

EGFR pathways

Mutations in the PIK3CA gene are associated with an increase in the severity of disease and worse clinical outcomes. PI3K/AKT signaling is involved in CRC, with a frequency of 15–20%, and can be mediated by miR-126, which mediates a reduction in p85β followed by a reduction in phosphorylated AKT levels in cancer cells, suggesting a deficiency in PI3K signaling [111,112]. Src inhibition reduces the interaction between Src and p85, subsequently decreasing Akt-dependent signaling. It was found that mutations in the miR-520a- and miR-525a-binding sites in the 3′UTR of PIK3CA could improve the sensitivity of CRC cell lines to an inhibitor of Akt-dependent signaling, saracatinib [113]. KRAS mutations are present in 30–60% of all CRC cases and mediate primary resistance to anti-EGFR-targeted therapy [114]. Some miRNAs, such as let-7 [115], miR-18a* [116], miR-30b [117], miR-143 [118] and miR-145 [119], have been shown to be tumor suppressors (since they inhibit KRAS expression) and potential biomarkers for predicting favorable responses to anti-EGFR therapy.

TGF-β signaling pathway

The transforming growth factor-beta (TGF-β) superfamily plays key roles in tissue maintenance, particularly in the context of inflammation and tumorigenesis, by modulating cell growth, differentiation and apoptosis. It was estimated that 30% of CRC cases showed mutations in the TGF-β type 2 receptor (TGF-βR2) [120,121]. TGF-β is activated by a protease and binds to its receptors (TGF-βR type I and II), mediating the triggering of its pathway through the phosphorylation of Smad. This complex is further translocated to the nucleus, and it modulates the expression of transcriptional factors (Snail, ZEB and Twist). Mismatch repair deficiency influences TGF-β receptor type 2 (TGFBR2) mutations and generates colorectal cancers (CRCs) with microsatellite instability, which is correlated with better survival rates. On the other hand, the loss of SMAD4, a transcription factor involved in the signaling of the TGF-β superfamily, promotes tumor progression and SMAD4-deficient CRC, and is linked to short survival for patients [122]. Several miRNAs, such as miR-21, miR-301a and miR-106a, have been reported to target the TGF-β/Smad signaling pathway, inducing stemness or cancer invasion and migration in CRC [123,124,125]. Conversely, the downregulation of the expression of miR-25 in CRC cell lines can promote SMAD7, an inhibitor of the TGF-β signaling pathway, enhancing cancer proliferation and the migratory ability of tumor cells [126].

Epithelial-to-mesenchymal transition (EMT)

Some miRNAs have been described as key regulators of this EMT process in CRC. The expression of the miR-34 family is promoted by p53, and the loss of miR-34a expression has been associated with EMT through the induction of the EMT transcription factor Snail [127]. MiR-29c has been demonstrated to be downregulated in metastatic CRC, and it was correlated with significantly lower patient survival. Importantly, the forced hyperexpression of miR-29c inhibits cell migration and invasion in vitro and metastasis in vivo, which is associated with its suppression of the AKT/GSK3β/β-catenin and ERK/GSK3β/β-catenin pathways [128]. A higher level of miR-200c was observed in liver metastasis and is specifically associated with the hypomethylation of the miRNA gene’s promoter [129]. The Wnt/β-catenin pathway was also shown to activate miR-150, which targets EMT in CRC by inhibiting CREB signaling [130].

### 4.2.1. MiRNAs as Potential Biomarkers in CRC

Tumor tissue from the surgical resection of primary tumors or metastasis can be an important source for finding CRC-related miRNAs. Various miRNAs have been reported to be upregulated or downregulated in CRC tumor specimens from patients, which suggests their association with the prognosis and response to the anticancer drugs of the patients. It is known that oncogenic miRNAs (oncomiRs) and tumor suppressor miRNAs have different expression during the development and progression of CRC.

miRNAs as prognostic biomarkers for CRC

There are some tissue-specific signatures of miRNAs that have been reported as prognostic biomarkers, but more work is necessary to establish the value of miRNAs as prognostic tools for guiding the treatment of CRC patients. Many miRNAs have been found to be upregulated or downregulated in CRC cell lines in tumor specimens from patients, suggesting their association with patients’ prognosis and the prediction of the response to anticancer drugs. Oncogenic miRNAs and tumor suppressor miRNAs are differentially expressed during the development and progression of CRC, and this miRNA expression in CRC is regulated in a stage-specific manner [114]. Therefore, the order of miRNA regulation during CRC progression may suggest the disease prognosis and predict the treatment response. The miR-17/92 cluster (also known as ‘oncomiR-1′) is very often amplified in CRC, and its members (miR-17, miR-18a, miR-20a, miR-19a, miR-19b-1 and miR-92a-1) are described as having oncogenic roles. The mir-17 and TNM staging systems are significant but independent prognostic biomarkers in CRC patients [131]. Other miRNAs were shown to regulate multiple targets to drive CRC progression (PTEN by miR-19a and miR-19b-1; BCL2L11 by miR-19b-1, miR-19a, miR-20a and miR-92; E2F1 by miR-20a and miR-17; TGF-β receptor 2 by miR-20a and miR-17 [132,133,134,135]). A high level of miR-21 in tumor specimens was shown to be correlated with distal metastasis in CRC patients [136]. Shybuia et al. reported on a cohort of 156 CRC patients with a high level of miR-21 and showed there was a correlation between liver metastasis and venous invasion [137]. A signature including increased miR-21, miR-93 and miR-103 was reported to be correlated with liver metastasis in CRC cases [138]. Additionally, miR-29a was reported to be a metastasis-promoting factor that targets the suppressor gene KLF4, upregulating matrix metalloproteinase 2 and downregulating E-cadherin [139]. A significant limitation of evaluating miRNAs in archival tumor tissues is heterogeneity (differences between the primary tumor and different metastatic sites). Therefore, detecting miRNA expression from human fluids may be preferable for predicting prognoses in the clinical setting. In this regard, there have been reports on the detection of three types of miRNA: circulating miRNAs (serum/plasma), fecal-based miRNAs and miRNAs in CRC-derived exosomes. Several circulating miRNAs have been evaluated for use as prognostic markers in CRC. A recent study revealed that a panel of miRNAs isolated from serum (let-7g, miR-21, miR-31, miR-203, miR-92a and miR-181b) was reported to have value as a prognostic marker in CRC, with 93% sensitivity and 91% specificity [140]. A high level of miR-141 in the plasma was reported as an independent prognostic factor in advanced CRC, predicting poor survival [141]. The plasma levels of miR-24, miR-320a and miR-423-5p were evaluated for the prediction of post-surgery metastasis in CRC patients, and high levels have promising potential to serve as novel biomarkers. Moreover, the sensitivities of miR-24, miR-320a and miR-423-5p were 77.78%, 90.74% and 88.89%, respectively [142]. MiRNAs are stable enough for detection in stool samples because they are protected in exosomes. Moreover, due to the direct contact of the stool with the lumen of the colon, molecular changes in CRC are easier to detect from fecal matter than from the blood [143]. Eight miRNAs (miR-9, miR-127-5p, miR-138, miR-29b, miR-143, miR-222, miR-146a and miR-938) had lower expression in the stools of patients with colon cancer, which was more pronounced in the later TNM stages [144]. miR-135b was found to be higher in CRC than in adenoma, with a specificity of 68% (Table 1).

Tumor-derived exosomes are known to be mediators of oncogenic transformation through the transfer of mRNAs, miRNAs and proteins during tumorigenesis [145]. The elevated expression of the exosomal miR-17-92a cluster was found to be correlated with recurrence in late-stage CRC patients. Additionally, an elevated exosomal level of miR-19a in the serum samples of CRC patients was reported to be correlated with poor prognosis [146]. A recent report has demonstrated an enrichment of the exosomal cargo of miR-328 from CRC patients’ plasma samples collected from mesenteric veins, when compared to peripheral veins, indicating a possible role of miR-328 in the development of liver metastases [147].

**Table 1 cancers-14-04834-t001:** miRNAs from tissue specimens, from free circulating/exosome cargo in the serum/plasma and from fecal samples suggested to have prognostic value in patients with metastatic colorectal cancer.

Type of Sample	miRNA	Method of Detection	Correlation with Clinical Outcome	Ref.
Tissue specimen	miR-15a/miR-16	qRT-PCR	Downregulation correlated with an advanced TNM stage, poor histologic grade, lymph node metastasis, and unfavorable OS and DFS	[148]
	miR-21	In situ hybridization	High expression correlated with poor survival and poor therapeutic outcomes; miR-21 regulates the expression of ITGb4, PDCD4, PTEN, SPRY2 and RECK	[149]
	miR-106a	qRT-PCR	Downregulation correlated with unfavorable OS	[150]
	miR-132	qRT-PCR	Downregulation correlated with unfavorable OS and the development of liver metastasis	[151]
	miR-150	qRT-PCR, In situ hybridization	Low expression associated with longer OS; high expression associated with unfavorable outcomes in patients treated with 5-FU-based chemotherapy	[152]
	miR-181a	qRT-PCR	Low expression associated with poor PFS in patients with wild KRAS treated with EGFR inhibitors	[153,154]
	miR-188-3p	Level 3 Illumina (from TCGA database)	High expression correlated with metastatic disease; lower OS and lower expression are correlated with BRAF status	[155]
	miR-195	qRT-PCR	Low expression associated with lymph node metastasis and an advanced tumor stage	[156]
	miR-199b	qRT-PCR and miRNA microarray	MiR-199b regulates the SIRT1/CREB/KISS1 signaling pathway, and high expression is associated with longer survival	[157]
	miR-215	qRT-PCR	High levels associated with poor overall survival	[158]
	miR-218	qRT-PCR	High miR-218 expression associated with the response to the first-line 5-FU treatment	[159]
Circulating miRNAs—serum/plasma	miR-21	qRT-PCR	Lower serum levels correlated with higher local recurrence	[160]
	miR-23b	qRT-PCR	Low plasma levels correlated with a shorter recurrence-free survival time and poorer overall survival	[161]
	miR-139-5p	qRT-PCR	High serum levels correlated with tumor recurrence and metastasis	[162]
	miR-141	qRT-PCR	High plasma levels correlated with poor prognosis	[141]
	miR-155	qRT-PCR	High serum levels correlated with tumor differentiation, regional and distant metastasis, and the clinical TNM stage	[163]
	miR-183	qRT-PCR	High plasma levels associated with regional and distant metastasis and tumor recurrence	[164]
	miR-203	qRT-PCR	High serum levels associated with short survival and metastasis	[165]
	miR-218	qRT-PCR	Low serum levels associated with the TNM stage, lymph node metastasis (LNM) and differentiation	[166]
	miR-221	qRT-PCR	High plasma level is a prognostic factor for poor overall survival	[167]
	miR-885-5p	qRT-PCR miRNA microarray	High serum levels correlated with poor prognosis, regional and distant metastasis	[168]
	miR-122	miRNA microarray	High plasma levels correlated with higher grading, and higher miR-200a, miR-200b and miR-200c levels were associated with increasing severity of the recurrence in metastatic CRC patients	[169]
	miR-200a			
	miR-200b			
	miR-200c			
Exosomes from serum/plasma	let-7a	qRT-PCR TaqMan	Upregulated serum levels are correlated with recurrence	[170]
	miR-21			
	miR-23a			
	miR-150			
	miR-223			
	miR-1246			
	miR-1229			
	miR-203	qRT-PCR	Upregulated serum levels are correlated with recurrence	[171]
	miR-548c-5p	qRT-PCR miRNA microarray	Downregulated serum level associated with increased risk of liver metastasis and later TNM stage	[172,173]
	miR-638			
	miR-5787			
	miR-8075			
	miR-68869-5p			
Fecal samples	miRNA signature	qRT-PCR	High miRNA signature associated with reduced DFS and OS	[174]
	miR-223/miR-222			
	miR-92a/miR-222			
	miR-16/miR-222			
	miR-20a/miR-222			
	miRNA panel	miRNA microarray, qRT-PCR	12 upregulated miRNAs (miR-7, miR-17, miR-20a, miR-21, miR-92a, miR-96, miR-106a, miR-134, miR-183, miR-196a, miR-199a-3p and miR-214) and 8 downregulated miRNAs (miR-9, miR-29b, miR-127-5p, miR-138, miR-143, miR-146a, miR-222 and miR-938) were found to differentiate TNM stages with high sensitivity and specificity	[142]
	12 upregulated			
	8 downregulated			

PFS—progression-free survival; OS—overall survival; DFS—disease-free survival.

### 4.2.2. MiRNAs for Predicting the Response to Systemic Therapy in mCRC

There is a strong research interest in designing a miRNA signature to predict a more personalized response to systemic anticancer treatments for CRC, with fewer adverse effects. The potential use of miRNAs as biomarkers in tissues or other human fluids to predict the response to drug therapies (capecitabine/oxaliplatin cytotoxic chemotherapy and antiangiogenic or anti-EGFR-targeted therapy) in CRC patients has been reported by various studies (Table 2). In metastatic CRC settings, the new options are to identify the most efficient regimens and to identify the order of the application of different regimes to prolong the survival of the patients. It was found that some miRNAs were linked to a favorable response to anti-EGFR therapy, and other miRNAs are correlated with poor prognoses with these therapies for mCRC. The most studied miRNA was let-7, which targets the mutant KRAS and is found in KRAS-mutated tumors, and a high level of let-7 is a predictive factor for a favorable response to anti-EGFR therapy [175]. MiR-7 targets EGFR, and a high expression of miR-7 is predictive of an unfavorable response to anti-EGFR therapy [176]. The efficacy of anti-EGFR monoclonal therapy is also associated with the KRAS/BRAF status and how KRAS/BRAF-wild-type patients respond to anti-EGFR therapy. In some studies on metastatic CRC patients, it was found that low expression of miR-529 and also of miR-181a was correlated with a lack of response to anti-EGFR therapy with cetuximab or panitumumab [177], and a predictive signature in tumors with wild-type or mutant KRAS for the cluster let-7c/miR-99a/miR-125b was described, where high levels were correlated with a longer PFS and OS in patients treated with anti-EGFR therapy [178]. The levels of miRNAs in human fluids (serum, plasma and circulating exosomes) or stools are also correlated with the response to therapy in mCRC. It was reported that a strong response to anti-VEGF (bevacizumab) therapy combined with cytotoxic therapy was related to decreased circulating levels of miR-126, while a high circulating level of miR-126 predicts a lack of response to bevacizumab–chemotherapy regimens [179]. A panel of five circulating miRNAs (miR-130, miR-145, miR-20a, miR-216 and miR-372) was reported to predict the response to classical regimens of chemotherapy, and it may help in the selection of these regimens [171]. Additionally, high levels of miR-155 and mir-345 are associated with poor responses to 5-FU/leucovorin/cetuximab and irinotecan/cetuximab in mCRC [180] (Table 2)

### 4.3. LncRNA

LncRNAs represent a type of ncRNA implicated in transcriptional processes and function in a similar way to positive or negative regulators. These small molecules can influence many biological processes such as cell proliferation, apoptosis, angiogenesis and stem cell self-renewal. Additionally, they can directly interact with mRNAs and regulatory protein complexes. In recent years, these molecules have been a major subject in cancer research because they can influence all the stages of cancer development [188,189]. In this review, we try to summarize the involvement of lncRNAs in metastatic CRC and some potentially therapeutic targets.

RP11 expression in CRC cells seems to correlate with lymph node metastasis and the advanced TNM stage, suggesting that this molecule can be a strong predictor of CRC metastasis and prognosis. Additionally, the upregulation of RP11 by m^6^A regulation can trigger the migration, invasion and EMT of CRC cells via the post-translational upregulation of the EMT-promoting TF Zeb1 [190].SATB2-AS1 is a colorectal-specific lncRNA expressed in colorectal tissues and CRC cells that inhibits tumor metastasis and regulates the immune response by activating SATB-2 in CRC. SATB2-AS1 downregulation seems to be due to DNA hypermethylation and histone H3K4me3 loss in the promoter region. Low levels of this lncRNA are correlated with the tumor invasion depth, lymph node metastasis and the TNM stage. Additionally, the gene signatures of the hallmark epithelial–mesenchymal transition, hallmark inflammatory response and hallmark interferon-gamma response were enriched in patients with low SATB2-AS1 expression. Overall, low SATB2-AS1 expression was associated with poor survival, and this study suggests that SATB2-AS1 and SATB2 may be novel biomarkers and promising therapeutic targets in CRC [191].LINC00659 expression in CRC is associated with poor prognosis. This study revealed higher levels of LINC00659 in CAF-exos than in NF-exos, which are transmitted to CRC cells and act through upregulating ANXA2 and increasing cell proliferation, migration and invasion [192].MALAT1 is another lncRNA that promotes CRC’s aggressiveness by regulating FUT4-associated fucosylation and the PI3K/Akt/mTOR pathway. In this study, we demonstrated that exosomes containing MALAT1 contributed to metastasis and the invasion of CRC cells via targeting miR-20b-5p, and targeting exosomal MALAT1 could attenuate the PI3K/AKT/mTOR pathway in CRC [193].

### 4.4. circRNA

As they have regulatory roles, circRNAs have maintained similar structures and roles during biological evolution [194]. CircRNAs are circular-shaped RNA transcripts, and they are produced through “back-splicing”: the ligation of the 3′ and 5′ ends of a linear RNA. This process can be induced by intron pairing, intron lariat formation and the dimerization of proteins. The result of “back-splicing” is a covalent loop. CircRNAs are involved in miRNA sponging, and they mediate the up- or downregulation of the expression of one or multiple targets of a miRNA. Furthermore, circRNAs can influence the translation of ceRNA transcripts [195]. Other identified roles of circRNAs include regulating proteins’ interactions and their ability to change their function and be translated into proteins, and the “back-splicing” of a pre-mRNA can interact with mature mRNA production [196]. CircRNAs are known to play critical roles in a cell’s cytoplasm and its protein interactions [197]. The intracellular circRNA level is mainly modulated through exosome removal [198]. All ncRNAs, including circRNAs, can leave the cell and interact with other cells through bodily fluids. They reach the target cells via a hormone-like mechanism that includes autocrine (a cell signals to itself), paracrine (induces changes in nearby cells) and endocrine (signals are carried to target cells in distant parts of the body) communication [199]. Exosomes (diameter of 30–160 nm), as a type of extracellular vesicles, can mediate the communication between two cells. This pathway is extremely controlled and well-studied. They are released into the bloodstream attached to protein and lipid complexes or inside extracellular vesicles and are, thus, protected against enzymatic activity such as that of exonucleases and ribonucleases. Thus, circRNAs are stable and have sufficient half-lives in body fluids. Exosomes contain different types of genetic material such as RNA (non-coding RNAs and messenger RNAs) and DNA, and other molecules such as proteins and lipids. Their surfaces contain proteins including major histocompatibility complex I and II, integrins, CD9, CD63 and CD81, and they facilitate the delivery of exosomes to target cells [200]. Because it reflects tumor features, the circRNA information that can be found in exosomes is specific. CircRNAs are also involved in tumoral anchorage-independent growth, invasion and the migration of colon cancer. Having the necessary characteristics to become an ideal biomarker for cancer diagnosis and prognosis, circRNA is tissue-specific and shows stable levels in bodily fluids such as tissue samples, saliva and blood [201].

#### 4.4.1. Candidate Prognostic Biomarkers in Metastatic CRC

Following changes in the expression of cell-junction proteins (E-cadherin) and mesenchymal proteins (N-cadherin, fibronectin and vimentin), epithelial cells lose their polarity and anchorage and become mobile. This process, known as EMT, is associated with metastasis and is activated through the modulation of signaling pathways by transcription factors such as ZEB1 [202]. CircRNAs also act along with transcription factors, and together, they regulate EMT. The increased level of hsa_circ_0001178 found in CRC cells increases ZEB1 expression, and, in addition to sponging miR-382/587/616, this increases N-cadherin expression [203]. Circ_ABCC1 and hsa_circ_0005075, which are blood- and tissue-based, modulate the Wnt/β-catenin pathway and regulate EMT. The alteration of their expression is associated with invasiveness and distant metastases [204,205,206]. CircCCDC66 is a well-known circRNA that can sponge numerous miRNAs that target key oncogenes in CRC. Elevated circCCDC66 expression leads to the derepression of certain tumor suppressor miRNAs (that target MYC, EZH2, YAP1 and DNMT3B) [207]. CircCCDC66 is a potential prognostic biomarker for metastatic CRC [207] (Table 3).

#### 4.4.2. Candidate Predictive Biomarkers in Metastatic CRC

Chemotherapy still plays a crucial role and can palliate the symptoms and prolong the lives of advanced CRC patients. Chemoresistance occurs through various mechanisms in almost all patients and is associated with a poor prognosis. The main mechanisms of drug resistance include: (1). the excessive expression of ATP-binding cassette (ABC) transporters in tumor cells causing the drug’s excretion from the cell and, thus, a dramatic decrease in its intracellular concentration and therapeutic effect; (2). an increase in the expression of antiapoptotic genes or the repression of the expression of tumor suppressor genes [241]; (3). increased DNA repair capacity, decreasing the effectiveness of the drug’s destruction of the structure of DNA, and 4; the tumor microenvironment being characterized by the presence of hypoxia and acidosis, but also fibroblasts and the neoformation of vessels, protecting tumor cells from the action of the immune system and drugs [242]. Some studies have analyzed the role of circRNA in the occurrence of therapeutic resistance and have demonstrated a correlation between increased levels of their expression in neoplastic cells and resistance to oxaliplatin, 5-FU or doxorubicin.

Laboratory-observed cell lines (HCT116-P) from patients diagnosed with CRC treated with FOLFOX/XELOX protocols and evaluated according to the RECIST 1.1 criteria were further exposed to ten cycles (with one cycle lasting 14 days) of 5 μM 5-FU and 0.625 μM oxaliplatin for 48 h. The surviving cells (HCT116-R) were maintained in a drug-free medium for 12 days and then in a medium with low concentrations of 5-FU and oxaliplatin (3125 μM of 5-FU + 0.625 μM of oxaliplatin). 

Correlated with the higher concentration of circ_0000338 in HCT116-R exosomes, HCT116-R cells are 5.78 times more resistant to 5-FU and 2.58 times more resistant to oxaliplatin compared to HCT116-P cells. HCT116-R are 5.78 times more resistant to 5-FU and 2.58 times more resistant to oxaliplatin compared to HCT116-P cells [243]. 

The circRNA profile of chemotherapy-resistant cells is different from that of chemotherapy-sensitive cells. Thus, an elevated level of hsa_circ_32883 was observed in HCT116-R cells, correlating with the increased resistance to FOLFOX therapy of these cells [244]. Other circRNAs that promote 5-FU resistance are circ_0007031 [76], circ_0000504 [76], circ-PRKDC [245] and circ0007006 [246]. By contrast, the overexpression of circDDX17 is correlated with the increased 5-FU sensitivity of neoplastic cells [247].

Via exosomes, circ_0005963 is transferred from oxaliplatin-resistant to sensitive cells, producing changes in glycolysis and PMK2 expression and eventually inducing oxaliplatin resistance in cells that were initially sensitive to oxaliplatin [248].

Hsa_circ_0079662 is overexpressed in cell lines exposed to different concentrations of oxaliplatin that have acquired resistance to its action (HT29-LOHP, HCT116-LOHP and HCT8-LOHP) [249]. Following the serial administration of known oxaliplatin concentrations to the HCT116 and HT-29 cell lines, a dose-dependent increase in circCCDC66 is observed. CircCCDC66 expression is necessary for cell survival during the cellular stress caused by oxaliplatin administration, but also for the appearance of the treatment-resistant population [250]. Circ_001680 can promote the irinotecan therapeutic resistance of CRC cells [251].

## 5. Metabolomics

The gut microbiota is composed of many microorganisms, including viruses, fungi, bacteria and archaea, that interact with the intestinal cells of the host and play multiple roles in gut physiology and pathology [252,253]. The highest density of bacterial cells per gram of content in the gastrointestinal (GI) tract is found in the colon (3 × 10^13^ cells/gram of content), in contrast with the stomach (ten cells/gram of content) or the small intestine (10^3^–10^7^ cells/gram of content), with important roles impacting nutrition and immunity. Changes in the microbiota equilibrium could cause or perpetuate the development and evolution of several diseases [254,255,256,257]. The intestinal microbiota is over 90% composed of the phyla Firmicutes and Bacteroidetes, along with Proteobacteria and Actinobacteria, and one of their functions is the production of metabolites such as vitamin B and K, as well as hormones and other essential bioactive compounds [258,259,260,261]. In addition, the gut microbiota is responsible for adequate enteral immune-system function and homeostasis and, regarding GI cancer, some microorganisms are connected with carcinogenesis through the modulation of inflammation and the production of toxic metabolites [256,258,262]. The strong link between microbiota alterations and the appearance or progression of colorectal cancer (CRC) is supported by the high load of microorganisms in the distal part of the GI tract and by the preponderance of CRC cases in the context of the digestive oncological pathology [261,263]. The term dysbiosis is defined by modifications in the protective roles of the microbiota due to changes in dietary habits or environment [264]. Dysbiosis could cause a disruption in the host and microbiota physiology that could result in the appearance of CRC [265]. The proposed pathophysiological mechanism for this interrelation is that dysbiosis causes inflammation that prevents the colonic epithelial cells from forming an efficient barrier against microorganisms; therefore, bacteria can easily invade the underlying tissue and promote tumorigenesis through further inflammation [261,266,267]. Scientific data indicate that species of the genera Escherichia, Enterococcus, Bacteroides and Clostridium could increase aberrant crypt foci and thus promote carcinogenesis [268,269]. Further scientific studies reveal the role of *Fusobacterium nucleatum*, *Helicobacter pylori*, *Streptococcus bovis*/*gallolyticus*, *Clostridium septicum* and *Bacteroides fragilis* as key factors in the carcinogenesis of CRC (Table 4) [270,271,272,273,274]. 

### 5.1. Gram-Negative Bacteria

*Fusobacterium nucleatum* is a Gram-negative, obligate anaerobic bacterium that causes an increase in the levels of polyamines (N1-acetylspermidine and N1,N12-diacetylspermine), especially in the tumoral microenvironment, and also increases the levels of proinflammatory cytokines (interleukin 17F, interleukin 21 and interleukin 22), leading to higher proliferation and invasiveness of CRC cells [275,276]. Polyamines are metabolites that induce oxidative stress and DNA damage that accelerate carcinogenesis in CRC [277]. A study by Ye et al. demonstrated a connection between *F. nucleatum* and increased levels of the oncogenic interleukin 17a (IL-17a) and tumor necrosis factor-α (TNF-α), and, furthermore, *F. nucleatum* inhibits T-cell and NK-cell antitumor activity [278,279].

*Escherichia coli* strains have previously been linked to intestinal inflammatory diseases and colon cancer [280]. *E. coli* strains are divided into the phylogenetic groups A, B1, B2 and D; the last two most frequently possess harmful virulence factors [281]. Some strains secrete metabolites that play important roles in CRC’s pathophysiology such as cycle-inhibiting factor (CIF), cytotoxic necrotizing factor (CNF), cytolethal distending toxin (CDT) and colibactin [261]. CIF, CDT and CNF are responsible for inducing cell-cycle arrest through different mechanisms, and mutations and could play a vital role in carcinogenesis [261,281]. Colibactin is a toxic compound secreted by the phylogenetic group B2 of *E. coli* that generates DNA damage, with the activation of the DNA damage checkpoint pathway and cell cycle arrest [282]. Scientific evidence has demonstrated a potent mutagenic function of colibactin in *E. coli*-infected cells in CRC [283].

*Helicobacter pylori* has a strain-dependent effect, and CagA-producing strains show more virulence [281]. *H. pylori* enhances gastrin production, which, in turn, increases antiapoptotic B cell lymphoma 2 protein (BCL-2) and disturbs acid production, with a negative impact on the intestinal epithelial cell barrier [284,285]. In addition, *H. pylori* stimulates the production of cytokines that are proinflammatory such as IL-1, IL-6, IL-8, interferon-γ (IFN-γ) and TNF-α [286].

Enterotoxigenic *Bacteroides fragilis* colonization was discovered in precancerous and cancerous lesions, highlighting the role of *B. fragilis* in the initial stages of carcinogenesis [261,286]. The oncogenic roles of *B. fragilis* are mediated through the secretion of enterotoxins responsible for polyamine catabolism and reactive oxygen species (ROS) production, the cleavage of E-cadherin and the overexpression of IL-17, which induce cell proliferation and inflammation [287,288]. 

### 5.2. Gram-Positive Bacteria

*Streptococcus bovis*/*gallolyticus* antigen is a potent activator of the cyclooxygenase 2 (COX-2) enzyme, which initiates angiogenesis and suppresses apoptosis [289]. Mucosa and stool samples of CRC patients demonstrated high numbers of *S. bovis*/*gallolyticus*, which were strongly linked to a high expression of the proinflammatory nuclear factor κB (NF-κB) and IL-8 mRNA [271]. The levels of *Enterococcus faecalis*, which are facultative anaerobic commensal bacteria, were found to be lower in the guts of healthy controls compared with those of colorectal patients [261,290]. *E. faecalis* induces DNA damage and chromosomal instability through the production of ROS such as extracellular superoxide and hydrogen peroxide [281,291]. *Clostridium septicum* is a spore-forming obligate anaerobic bacterium that is not found in the human digestive tract under normal conditions [261]. Its virulence resides in the production of α-toxin, which is hemolytic [292]. *C. septicum* activates proinflammatory TNF-α production and survives in the hypoxic tumor microenvironment [273,293]. In the case of CRC patients with *C. septicum* colonization and bacteremia, mortality rates were discovered to be higher [294].

### 5.3. Microbiota as Biomarkers in Colorectal Cancer

*Fusobacterium nucleatum* appears to be a fundamental marker for CRC, either quantified alone or combined with the colibactin-producing bacteria *Clostridium symbiosum* or *Clostridium hathewayi* [295,296,297]. Measuring the fecal levels of *F. nucleatum* can increase the sensitivity and specificity of fecal immunochemical testing (FIT) in detecting CRC, compared to FIT alone [295,298]. This association demonstrates the advantages of testing different complementary targets with the ultimate goal of reducing missed cancer cases [268]. *F. nucleatum* could also be an important prognostic marker in CRC, as multiple studies have highlighted an association between high tumor amounts of *F. nucleatum* and decreased survival in CRC [299,300,301]. Various studies have shown an association between oral microbiota, including *Streptococcus* and *Prevotella* spp., and CRC and could indicate a possible role for the oral microbiota in CRC prognosis [302,303]. Furthermore, a positive serological test for *S. gallolyticus* was associated with CRC development even 10 years after the test [304].

## 6. Artificial Intelligence Methods Used in mCRC

Since 2010, the use of AI in medical disease diagnosis and treatment has grown over the years [305,306]. AI techniques have been used with success in many contexts, including colon polyps, adenomas, colon cancer, ulcerative colitis and intestinal motor diseases. Although the application of AI to the diagnosis and treatment of CRC still lacks systematic research, the continuous development of AI applications in the medical field is an indication that AI will eventually be used for the diagnosis and therapy of CRC. 

A classification of AI applications for the identification of new prediction/prognosis biomarkers in mCRC is related to machine learning (ML) models that can be described according to the basic features: (1) support vector machines (SVMs) that are supervised learning models with associated learning algorithms that analyze data for classification and regression analysis, and; (2) the artificial neural networks (ANN) usually simply called neural networks (NNs) or neural nets, including convolutional neural network (CNN, or ConvNet), that can be defined as regularized versions of multilayer perceptrons.

ML is divided into supervised and unsupervised based on whether the training data is labeled or not [307].

### 6.1. AI Application for Developing Biomarkers in mCRC in Blood Tests and Other Tests 

Xu et al. [308] 2017, used an SVM system and identified a 15-gene signature (HES5, ZNF417, GLRA2, OR8D2, HOXA7, FABP6, MUSK, HTR6, GRIP2, KLRK1, VEGFA, AKAP12, RHEB, NCRNA00152, and PMEPA1) with differentially expressed genes (DEGs) as a predictor of recurrence risk and prognosis in mCRC patients. A new method called the “walking pathway” was designed by Kel et al., to search for potential rewiring mechanisms in cancer pathways due to changes in the DNA methylation status of regulatory gene regions (“epigenomic walking”). Researchers analyzed an extensive collection of complete genome gene-expression data (RNA-seq) and DNA methylation data of genomic CpG islands from a sample of the tumor and normal gut epithelial tissues of 300 patients with colorectal cancer using the web service “My Genome Enhancer” (MGE), gene regulation database TRANSFAC^®^, the signal transduction pathways database TRANSPATH^®^, and AI software [309].

An artificial neural network (CP-ANN) was developed by Zhang et al., to obtain higher sensitivity and lower cost for the detection of the BRAF gene mutation, where valine was substituted by glutamic acid at codon 600 (V600E) in CRC. The CP-ANN achieved a diagnostic sensitivity of 100%, specificity of 87.5%, and accuracy of 93.8% [310].

In 2014, Tutar considered that AI technology could be seen as a bridge to connect ncRNAs with tumor researchers [311]. In the beginning, the research was based on naive Bayes classifiers and on ANN algorithms. In the paper [312], the authors used an ANN algorithm to measure different expression profiles of microRNAs (miRNAs). They identified three miRNAs (miR-139-5p, miR-31 and miR-17-92) that could predict the tumor status of stage II CRC. Later, in 2015, Amirkhah proposed a miRNA-associated tumor prediction method based on naive Bayes classification, called CRCmiRTar [313]. The ShrinkBayes model has been demonstrated to have good predictive accuracy through studies with small sample sizes or complex designs [314]. Later research activities were related to using CNN-based methods. Xuan et al. proposed a dual-CNN-based prediction method for disease-related miRNAs that explores the deep features of miRNA similarities and disease similarities [315]. Afshar et al. screened four CRC-specific miRNAs from a database and accurately classified the sample data as cancerous and non-cancerous data using an ANN [316]. Using an SVM classification model, the results from testing on 297 patients was 85% [317]. Other researchers have used three datasets and CNN combined with adversarial training to minimize the specific features of the dataset [318]. Cao et al. proposed a multiple instance learning (MIL)-based, deep-learning pipeline, which made predictions at two levels: the patch-level and whole-slide-image (WSI) level [319]. Some experiments also used machine-learning (ML) and computer-vision (CV) techniques to identify the percentages of positive tumor cells within tumor areas for AREG and EREG [320]. A CNN model (VGG-19) was tested for classifying different types of pathologic CRC images [321]. In a review of AI methods used in medical applications targeting personalized therapies for cancer, it was shown that artificial neural networks (ANNs), logistic regression (LR) and support vector machines (SVMs) were the preferred models [322].

### 6.2. AI Application in the Personalization and Precision Treatment of mCRC

The IBM Corporation, together with the Memorial Sloan Kettering Cancer Center, developed a system called “Watson for Oncology” (WFO), an AI system that can assist the decision of personalized treatment in mCRC. This AI system can automatically extract medical characters from doctor records and translate them into a practical language for learning. WFO’s recommendations are approx. 90% consistent with the guidance of the multidisciplinary team (MTD), according to Dr. Anderson [323]. In a South-Korean trial, the concordance between WFO and MTD increased to 88,4% [324]. Additionally, WFO was experimented with in Tokyo University for gene-sequencing in cancer, significantly reducing wait time [325]. Keshava et al. developed an AI model that can identify different subpopulations with different responses to target inhibitors, being revealed more information about the mechanisms of resistance and pathway cross-talk [326]. 

AI has shown impressive performance for targeted drugs. Another AI system was designed by Ding et al. to screen molecular markers from proteomics and transcriptomics data. It predicted which protein products of the common up-regulated and down-regulated genes can be secreted into blood, urine, or saliva using SVM-based classified models. 

Three genes: ESM1, CTHRC1, and AZGP1 can be secreted into saliva, blood, and urine simultaneously and are predicted as candidate biomarkers for colorectal cancer. ESM1 promotes angiogenesis; CTHRC1 is a negative regulator of collagen matrix deposition and AZGP1 activates the lipid degradation in adipocytes [327]. Lee et al. [328] developed an AI model that predicts the protein-protein interactions of S100A9 with different drugs by applying machine learning classifiers on 2D-molecular descriptors. An efficient FASTCORMICS RNA-seq workflow to build 10,005 high-resolution metabolic models from the TCGA dataset was designed to capture metabolic rewiring strategies in cancer cells in mCRC patients. The metabolic model based on RNA-seq data successfully predicts drug targets and drugs not yet indicted for colorectal cancer. Was demonstrated that cancer-type and patient-specific drugs can be identified if the workflow is used together with machine learning, identifying different patient groups with different responses to drugs [329].

### 6.3. AI for Developing Biomarkers to Predict and Prognosticate the mCRC

Deep-learning models based on protein–protein interaction networks to diagnose CRC metastases by selecting more effective molecular markers and algorithm parameters in a two-stage model were designed. In the first stage, particle swarm optimization (PSO) and differential evolution (DE) is used to optimize parameters of the support vector machine recursive feature elimination algorithm, and a dynamic Bayesian network is then used to predict the temporal relationship between biomarkers across two-time points. Results show that 18 and 25 biomarkers selected by PSO and DE-based showed the same accuracy of 97.3% and 97.6% [330]. Another study combined the logistic regression model (LRM) with an ANN system to design a mixed prediction model in which the most effective parameters were selected by the LRM to build hybrid predictors of metastasis in mCRC [331]. A protein-protein interaction (PPI) network was designed for these DEGs to identify the differentially expressed genes (DEGs), and the SVM-classified gene signatures were identified. 

In total, 358 DEGs were identified by meta-analysis. Based on the SVM classification model, 40 signature genes were described to be related to the AMPK signaling pathway (e.g., CREB1), endoplasmic reticulum (e.g., SSR3), and ubiquitin-mediated proteolysis (e.g., FBXO2, CUL7 and UBE2D3) pathways. These genes have the potential to be used as biomarkers for the prognosis of metastatic CRC [332].

### 6.4. Implementation of the Selected Predictive Models

We experimented with different machine-learning algorithms. In order to exploit the capacity of machine-learning algorithms or algorithms that use neural networks, we used several colorectal cancer datasets such as “Metastatic Colorectal Cancer—MSKCC, Cancer Cell 2018”. This resource contains the data of 1134 patients aged between 20 and 80 years, distributed by age as shown in Figure 2.

The data for a patient are varied and contain the following: the gene, information about the protein change, the annotation, the mutation type, the allele frequency, etc. (see Figure 3).

Of these data, we used 90% for training and 10% for tests. The relevance of the features selected and the correlations between them are presented in Figure 4.

We implemented six algorithms: (1) the naive Bayes classifier (a probabilistic classifier) [333], (2) the random forest classifier (an ensemble learning method) [334], (3) the decision tree classifier (which creates the classification model by building a decision tree) [335], (4) gradient boosted trees (an ensemble of weak prediction models) [336], (5) logistic regression (models’ probability of output in terms of input) [337] and (6) SVM (supervised learning models with associated learning algorithms) [338]. In the implementation, we used our experience from implementing similar solutions for fake news identification [339] and for plant identification [340].

### 6.5. Predictive Model Mobile App

In recent years, many mobile applications have used the power of artificial intelligence models to perform classification or prediction. Android applications take data from the user, then send them to a background process to analyze them and use the model to give an answer to the user. In our case, after collecting the data from the user with the help of an Android application, they were sent to a web server in order to apply the currently developed AI algorithms to the data. The application uses the client–server architecture, where the client is the Android application and the server component is represented by services that access the model built with the help of artificial intelligence. 

We conducted some experiments in order to improve two aspects: (1) the quality of the used model, and (2) the speed of the model that provides answers to users. To date, we have obtained a pilot solution of quite good quality that offers solutions in a very short amount of time, i.e., close to real time. This solution will be further updated as the AI algorithms are upgraded to a newer version.

#### 6.5.1. Experiments

As previously stated, in our experiments, we implemented six algorithms: (1) the naive Bayes classifier, (2) the random forest classifier, (3) the decision tree classifier, (4) gradient boosted trees, (5) logistic regression and, (6) SVM. The features used can be seen in Figure 3. Considering the values in the dataset were textual and the algorithms use a numerical input, a dictionary solution was adopted in order to translate the text into numbers. For example, “patient_id” 0000119 is of “age” 67, and the “chemo_exposure” value is in the lungs with the “first_site_of_metastasis” being in the liver and peritoneum; the translated values for the algorithms are “patient_id”: 0000119, “age”: 67, “chemo_exposure”: 1 (1 being lung), and “first_site_of_metastasis”: 2, 3 (2 being liver and 3 being peritoneum). Aside from this translation, the removal of negative values in the dataset was necessary in order to adhere to the prerequisites of the machine-learning algorithms. The algorithms implemented use the PySpark machine-learning library as well as sklearn and pandas. The classified column is the “living_status”, meaning that the algorithms try to classify whether the respective patient is alive or not.

#### 6.5.2. Naive Bayes

The settings for this algorithm were smoothing = 1.0 and model = “multinomial”. Table 5 shows the classification report, and Table 6 shows the confusion matrix. In this case, the obtained accuracy had the lowest value from all the implemented algorithms.

#### 6.5.3. Random Forest

The settings for this algorithm are maxDepth = 5, maxBins = 32, minInstancesPerNode = 1, minInfoGain = 0.0, maxMemoryInMB = 256, impurity = “gini” and numTrees = 20. Table 7 shows the classification report, and Table 8 shows the confusion matrix. The obtained accuracy is very close to the highest value from all the implemented algorithms.

#### 6.5.4. Decision Tree

The settings for this algorithm are maxDepth = 5, maxBins = 32, minInstancesPerNode = 1, minInfoGain = 0.0 and maxMemoryInMB = 256. Table 9 shows the classification report, and Table 10 shows the confusion matrix. The obtained accuracy in this case had the highest value from all the implemented algorithms.

#### 6.5.5. Gradient Boosted Trees

The settings for this algorithm were default, maxDepth = 5, maxBins = 32, minInstancesPerNode = 1, minInfoGain = 0.0, maxMemoryInMB = 256, cacheNodeIds = False, checkpointInterval = 10, lossType = “logistic”, maxIter = 20, stepSize = 0.1, seed = None, subsamplingRate = 1.0, impurity = “variance”, featureSubsetStrategy = “all”, validationTol = 0.01, validationIndicatorCol = None, leafCol = “”, minWeightFractionPerNode = 0.0, and weightCol = None. Table 11 shows the classification report, and Table 12 shows the confusion matrix. The obtained accuracy was the highest value from all the implemented algorithms.

#### 6.5.6. Logistic Regression

The settings for this algorithm are default, maxIter = 100, regParam = 0.0, elasticNetParam = 0.0, tol = 1 × 10^−6^, fitIntercept = True, threshold = 0.5, thresholds = None, probabilityCol = “probability”, rawPredictionCol = “rawPrediction”, standardization = True, weightCol = None, aggregationDepth = 2, family = “auto”, lowerBoundsOnCoefficients = None, upperBoundsOnCoefficients = None, lowerBoundsOnIntercepts = None, and upperBoundsOnIntercepts = None. Table 13 shows the classification report, and Table 14 shows the confusion matrix. The obtained accuracy is close to the highest value from all the implemented algorithms.

#### 6.5.7. SVM

The settings for this algorithm were maxIter = 100, regParam = 0.0, tol = 1 × 10^−6^, threshold = 0.0, and aggregationDepth = 2. Table 15 shows the classification report, and Table 16 shows the confusion matrix. The obtained accuracy is very close to the highest value from all the implemented algorithms.

## 7. Discussion

Although research has led to major advantages through new information on the driver genes responsible for carcinogenesis and metastasis, the transcriptional and epigenetic aberrations in this malignancy that influence many central signaling pathways have recently aroused increased interest. The ability of treatments to alter several different molecular pathways may have crucial implications for their efficacy. Epigenetic changes play a decisive role in the epithelial-to-mesenchymal transition (EMT), which is an essential phenotype for metastasis and includes DNA methylation, non-coding RNAs (ncRNAs), and N6-methyladenosine (m6A) RNA, which are valuable biomarkers in CRCs. For ncRNAs, a “molecular sponge effect” was described between long non-coding RNAs (lncRNAs), circular RNAs (circRNAs) and microRNAs (miRNAs). 

Many studies have focused on miRNAs as biomarkers for the following reasons: miRNAs represent small molecules, and they can remain intact in vivo or in vitro for a long time; extracellular circulating miRNAs are stable for at least one month [341]; miRNAs have been detected in various tissues, and it has been demonstrated that they participate in a variety of physiological and pathological processes [342]; in many studies, aberrant miRNA expression was found to also play crucial roles in the response to anticancer drugs [343], and; miRNAs change during treatment with anticancer drugs [344]. The implementation of AI in the identification of prognostic and predictive biomarkers in mCRC represents a new step in the progress of biomarker identification. NcRNAs play crucial roles in modulating the resistance to molecular therapies in CRC, including oncogenes and tumor suppressors such as miRNAs, lncRNAs and circRNAs, based on miRNA–mRNA, lncRNA–miRNA–mRNA or circRNA–miRNA–mRNA regulatory networks through the EGFR signaling pathway, the RAS signaling pathway and the PI3K/AKT signaling pathway (Figure 5). In this context, ncRNAs may function as novel biomarkers for predicting the prognosis with, efficacy of and resistance to anti-EGFR therapy in CRC. Further studies need to investigate new therapeutic strategies based on ncRNA regulatory networks. Due to the complexity of the molecular mechanisms and the multitude of genetic and epigenetic actors involved in the natural history of mCRC and its treatment, the implementation of AI in the identification of prognostic and predictive biomarkers in mCRC represents a new step towards the progress of biomarker identification. Machine-learning (ML) and deep-learning (DL) strategies are extremely powerful tools with which to analyze and interpret genomic and biological data that are useful for providing prognostic and predictive information. DL methods can be applied, together with imaging techniques, to correlate CRC molecular features with morphological markers. Two studies showed that the CRC MSI status and CMS transcriptional classification were more predictive than classical molecular profiling using DL combined with infrared imaging and hematoxylin and eosin (H&E)-stained tissue sections [345,346]. A network-based ML model was used to evaluate pharmacogenomic data derived from CRC organoids, identifying drug-response biomarkers linked to the 5-FU response [347]. In our study, we tried to design a predictive mobile app model and our experiments aimed to improve the quality of the model used and the speed of the model that provides answers to users. We obtained a pilot solution that is close to real time in performance and that should be updated when the AI algorithms are upgraded to newer versions.

## 8. Conclusions

In recent years, artificial intelligence has been used more and more to identify colorectal cancer, with promising results. As observed in the experiments, AI algorithms help to correctly classify patients with good results and can support physicians in their daily activities.

## Figures and Tables

**Figure 1 cancers-14-04834-f001:**
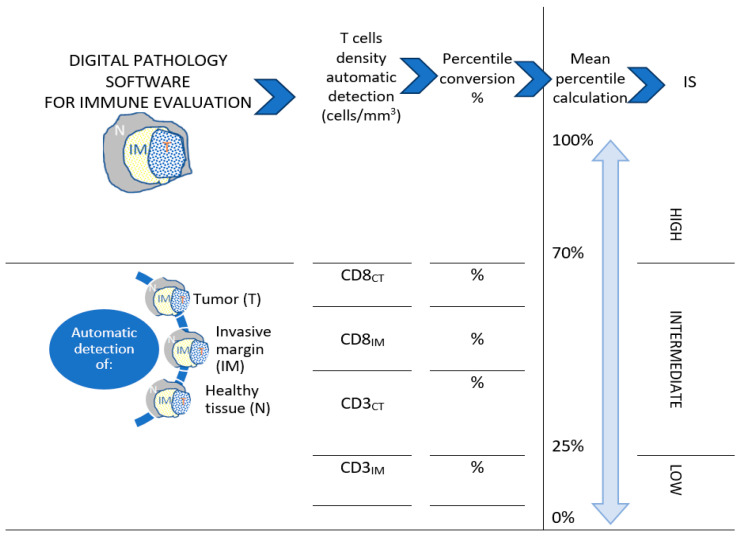
Diagrammatic presentation of immunoscore (IS) determination.

**Figure 2 cancers-14-04834-f002:**
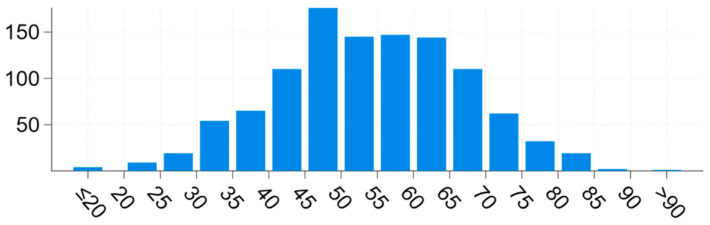
Patients’ distribution by age.

**Figure 3 cancers-14-04834-f003:**
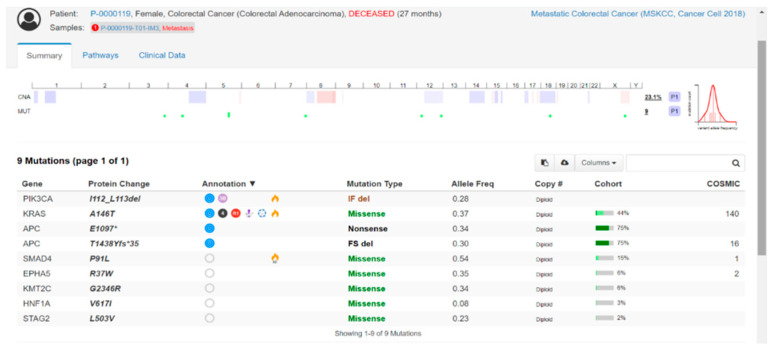
Information about a patient.

**Figure 4 cancers-14-04834-f004:**
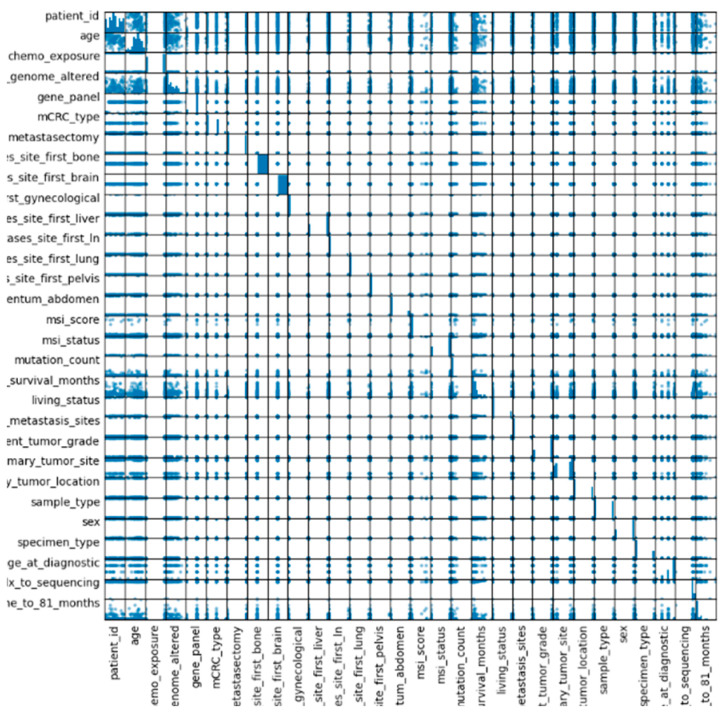
Correlation between the considered features.

**Figure 5 cancers-14-04834-f005:**
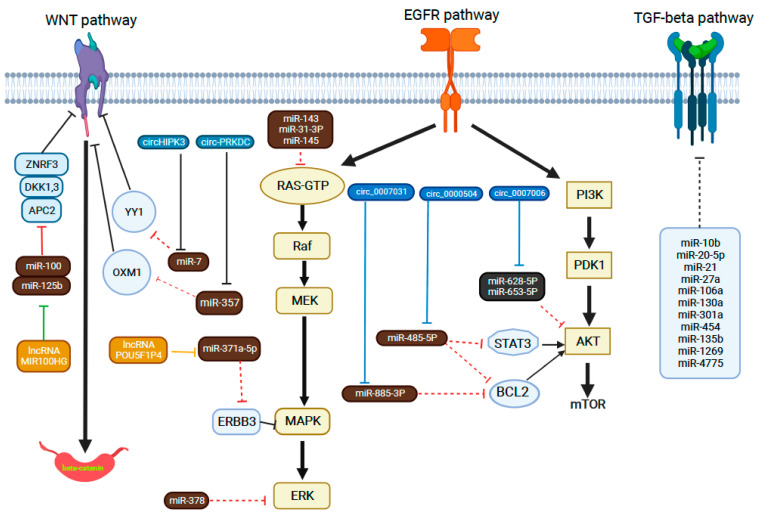
Mechanisms by which ncRNAs (circRNAs, lncRNAs, miRNAs) target the main pathways involved in the pathogenesis of CRC (created with Biorender.com).

**Table 2 cancers-14-04834-t002:** MiRNAs from plasma, serum and exosomes for predicting the response to systemic therapy in mCRC.

miRNA	Expression That Suggests Inadequate Response	Treatment Regimen	Molecular Mechanism	Detection Method	Ref.
Tissue specimen					
let-7	Low	Cetuximab–irinotecan	Let-7 targets KRAS and improves survival only withKRAS mutations	qRT-PCR	[175]
miR-7	Low	Cetuximab	MiR-7 suppresses EGFR	qRT-PCR	[176]
miR-31*	High	Anti-EGFR	MiR-31* targets the mRNA levels of SLC26A3 and ATN1	qRT-PCR	[177]
miR-143	High	Capecitabine, oxaliplatin and anti-EGFR	Modulation of KRAS by miR-143	Microarray, qRT-PCR	[181]
miR-145	Low	Cetuximab	Overexpression of cetuximab-mediated antibody-dependent cellular cytotoxicity	qRT-PCR	[182]
miR-146b-3p	High	Cetuximab	SP1/miR-146b-3p/FAM107A axis	qRT-PCR	[183,184]
miR-181a	Low	Anti-EGFR	miR-181 expression activated Wnt/β-catenin signaling	qRT-PCR	[181]
miR-200b	Low	Anti-EGFR	MiR-200b inhibits ERRFI mRNA in KRAS mutations	Microarray, qRT-PCR	[181]
miR-455-5p	High	Capecitabine, oxaliplatin and bevacizumab	MiR-455-5p downregulates the expression of PIK3R1	qRT-PCR, ISH	[185]
miR-592	Low	Anti-EGFR	MiR-592 targets the mTOR and FOXO signaling pathways	Microarray, qRT-PCR	[177]
miR-664-3p	Low	Capecitabine, oxaliplatin and bevacizumab	MiR-664-3p targets angiogenesis	qRT-PCR, ISH	[185]
signature let-7c, miR-99a and miR-125b	Low	Anti-EGFR	In wild-type KRAS	Microarray, qRT-PCR	[178]
miR-320e	High	5-FU	MiR-320e targets PP2R2C, IRF6, ONECUT2, CMCL1 and CPEB genes	Microarray	[186]
Serum/plasma					
miR-19a	High	FOLFOX	Targeted tumor suppressor genes, including E2F1, CDKN1A, PTEN, BCL2L11 and c-Myc	Microarray, qRT-PCR	[187]
miR-126	High	Cetuximab			[179]
miR-155	High	Leucovorin, 5-FU and cetuximab		qRT-PCR	[180]
miR-345	High	Cetuximab and irinotecan	EGFR inhibits miR-345 maturation	Microarray, qRT-PCR TaqMan	[181,182]
miR-106a, miR-484 and miR-130b miR-27b, miR-148a and miR-326	High	5-FU and oxaliplatin	Oncogenic miRNAs upregulated in metastatic disease	qRT-PCR	[183]
Exosomes					
Panel miR-100, miR-92a, miR-16, miR-30e, miR-144-5p and let-7i	Low	Oxaliplatin	Targets of ATG4B, BCL2, CCNJ and FUBP1	qRT-PCR	[184]
miR-92a-3p	High	5-FU and oxaliplatin	CAF-derived exosomes transfer miR-92a-3p, enhancing cell stemness, EMT, metastasis and chemoresistance	qRT-PCR	[185]
Panel miR-21-5p, miR-1246, miR-1229-5p, miR-135b, miR-425 and miR-96-5p	High	5-FU and oxaliplatin	Targets of the PI3K–Akt pathway, FOXO pathway and autophagy pathway	qRT-PCR	[186]
miR-125b	High	mFOLFOX6	Exosomal miR-125b has been correlated with chemoresistance	qRT-PCR	[187]

**Table 3 cancers-14-04834-t003:** Proposed prognostic biomarkers in metastatic colorectal cancer.

	circRNA	Blood/Tissue-Based	CircRNA’s Expression Level	Target Pathway/ Target miRNA	Biological Function
1	circ_0122319, circ_0087391, circ_0079480 [208]	Tissue	Increased	-	Promotes CRC metastasis
2	circ_ABCC1 [204]	Blood (plasma)	Increased	Wnt/β-catenin pathway	Promotes an advanced CRC stage with the involvement of the lymph node and distant organs
3	circ-0104631 [209]	Tissue	Increased	-	Promotes lymph node and distant metastasis
4	circCAMSAP1 [210]	Tissue	Increased	MiR-328-5p	Promotes an advanced TNM stage
5	circCDC66 [207]	Tissue	Increased	-	Promotes cancer cell proliferation, migration and metastasis
6	circCSNK1G1 [211]	Tissue	Increased	MiR-455-3p	Promotes aggressive cell proliferation, migration and distant metastasis
7	circFADS2 [212]	Tissue	Increased	-	Regulates cancer cell proliferation, invasion, EMT and metastasis
8	circ-FBXW7 [213]	Tissue	Decreased	NEK2, mTOR and PTEN signaling pathways	Controls tumor cell metastasis, stress response and immune functions
9	circHIPK3 [214]	Tissue	Increased	MiR-7	Promotes an advanced TNM stage
10	circHUWE1 [215]	Tissue	Increased	MiR-486	Promotes lymphovascular invasion, lymph node metastasis and distant metastasis
11	circ-ITGA7 [216]	Tissue	Decreased	Suppressing RREB1 via Ras pathway	Promotes lymph node metastasis, distant metastasis and an advanced TNM stage
12	circLONP2 [217]	Tissue	Increased	MiR-17	Promotes CRC metastasis
13	circMBOAT2 [218]	Blood	Increased	MiR-519d-3p	Promotes cell proliferation, invasion and metastasis
14	circ-NSD2 [219]	Tissue	increased	MiR-199b-5p/DDR1/JAG1	Promotes the migration, invasion and metastasis of CRC cells
15	circ-NSUN2 [220]	Tissue	Increased	IGF2BP2/HMGA2	Promotes CRC metastasis
16	circPPP1R12A [221]	Tissue	Increased	Activating Hippo-YAP signaling pathway	Promotes the proliferation and metastasis of cancer cells
17	circ-PVT1 [222]	Tissue	Increased	MiR-145	Promotes CRC liver metastasis
18	circRNA_100290 [223]	Tissue	Increased	MiR-516b	Promotes cell growth and metastasis in CRC, and suppresses apoptosis
19	circRNA_101951 [224]	Tissue	Increased	KIF3A-mediated EMT	Promotes colon cancer growth and metastasis
20	circVAPA [225]	Tissue	Increased	MiR-101	Promotes lymphovascular invasion, lymph node metastasis and distant metastasis
21	ciRS-7—A [226]	Tissue	Increased	MiR-7 a	Promotes lymph node and distant metastasis
22	has_circ_0055625 [227]	Tissue	Increased	MiR-106b-5p	Promotes mCRC development
23	hsa_circ_ 0000372 [228]	Tissue	Decreased	MiR-101-3p, miR-495, miR-485-5p	Promotes cancer progression
24	hsa_circ_0000567 [229]	Tissue	Decreased	-	Promotes cancer-cell proliferation and metastasis
25	hsa_circ_0001178 [203]	Tissue	Increased	MiR-382/587/616/ZEB1	Promotes colon cancer growth and metastasis
26	hsa_circ_0004831 [230]	Blood	Increased	MiR-4326	Promotes advanced CRC evolution
27	hsa_circ_0005075 [205,206]	Tissue	Increased	Wnt/β-catenin pathway	Promotes CRC metastasis
28	hsa_circ_0007534 [231,232]	Blood	Increased	-	Promotes progression to metastatic stage
29	hsa_circ_0014717 [233]	Tissue and plasma	Decreased	Upregulates the expression of cell-cycle-inhibitory protein p16	Promotes lymph node metastasis and distant metastasis
30	hsa_circ_0026416 [234]	Tissue and plasma	Increased	MiR-346/NFIB	Promotes colon cancer growth and distal metastasis
31	hsa_circ_0079993 [235]	Tissue	Increased	MiR-203a-3p.1	Promotes CRC metastasis
32	hsa_circ_0136666 [236]	Tissue	Increased	MiR-383	Promotes metastasis in the lymph nodes and distant metastasis
33	hsa_circ_100876 [237]	Tissue	Increased	MiR-516b	Promotes metastasis in the lymph nodes and distant metastasis
34	hsa_circ_101555 [238]	Tissue	Increased	MiR-597-5p	Promotes metastasis in the lymph nodes and distant metastasis
35	hsa_circRNA_002144 [239]	Tissue and plasma	Increased	MiR-615-5p/LARP1/mTOR	Promotes metastasis in the lymph nodes and distant metastasis
41	hsa_circRNA_102209 [240]	Tissue	Increased	MiR-761/RIN1 axis	Promotes colon cancer growth and distal metastasis

**Table 4 cancers-14-04834-t004:** Summary of pathogen mechanisms implicated in carcinogenesis.

Pathogen	Mechanism Implicated in Carcinogenesis	Reference
*Fusobacterium nucleatum*	Increased levels of polyamines Increased levels of proinflammatory cytokines	[275,276]
*Escherichia coli*	Secretion of CIF, CDT, CNF and colibactin Induction of cell cycle arrest	[261,281,282]
*Helicobacter pylori*	Increase in antiapoptotic B cell lymphoma 2 protein (BCL-2) levels Disturbance of gastric acid production Increase in proinflammatory cytokine levels	[284,285,286]
*Bacteroides fragilis*	Secretion of enterotoxins Cleavage of E-cadherin Overexpression of IL-17	[277,288]
*Streptococcus bovis*/*gallolyticus*	Activator of COX-2 Overexpression of NF-κB mRNA Overexpression of IL-8 mRNA	[271,289]
*Enterococcus faecalis*	ROS production DNS damage Chromosomal instability	[281,291]
*Clostridium septicum*	Hemolytic α-toxin production TNF-α production	[273,292,293]

**Table 5 cancers-14-04834-t005:** Naive Bayes classification report.

	Precision	Recall	F1-Score	Support
0	0.80	0.64	0.71	677
1	0.48	0.68	0.56	328
Accuracy			0.65	1005
Macro avg.	0.64	0.66	0.64	1005
Weighted avg.	0.70	0.65	0.66	1005

**Table 6 cancers-14-04834-t006:** Naive Bayes confusion matrix.

	Predicted Yes	Predicted No
Actual Yes	434	243
Actual No	106	222

**Table 7 cancers-14-04834-t007:** Random forest classification report.

	Precision	Recall	F1-Score	Support
0	1.0	1.0	1.0	677
1	1.0	0.99	1.0	328
Accuracy			1.0	1005
Macro avg.	1.0	1.0	1.0	1005
Weighted avg.	1.0	1.0	1.0	1005

**Table 8 cancers-14-04834-t008:** Random forest confusion matrix.

	Predicted Yes	Predicted No
Actual Yes	677	0
Actual No	3	325

**Table 9 cancers-14-04834-t009:** Decision tree classification report.

	Precision	Recall	F1-Score	Support
0	1.0	1.0	1.0	677
1	1.0	1.0	1.0	328
Accuracy			1.0	1005
Macro avg.	1.0	1.0	1.0	1005
Weighted avg.	1.0	1.0	1.0	1005

**Table 10 cancers-14-04834-t010:** Decision tree confusion matrix.

	Predicted Yes	Predicted No
Actual Yes	677	0
Actual No	0	328

**Table 11 cancers-14-04834-t011:** Gradient boosted tree classification report.

	Precision	Recall	F1-Score	Support
0	1.0	1.0	1.0	677
1	1.0	1.0	1.0	328
Accuracy			1.0	1005
Macro avg.	1.0	1.0	1.0	1005
Weighted avg.	1.0	1.0	1.0	1005

**Table 12 cancers-14-04834-t012:** Gradient boosted tree confusion matrix.

	Predicted Yes	Predicted No
Actual Yes	677	0
Actual No	0	328

**Table 13 cancers-14-04834-t013:** Logistic regression classification report.

	Precision	Recall	F1-Score	Support
0	0.97	0.99	0.98	677
1	0.97	0.94	0.96	328
Accuracy			0.97	1005
Macro avg.	0.97	0.96	0.97	1005
Weighted avg.	0.97	0.97	0.97	1005

**Table 14 cancers-14-04834-t014:** Logistic regression confusion matrix.

	Predicted Yes	Predicted No
Actual Yes	667	10
Actual No	19	309

**Table 15 cancers-14-04834-t015:** SVM classification report.

	Precision	Recall	F1-Score	Support
0	1.0	1.0	1.0	677
1	1.0	1.0	1.0	328
Accuracy			1.0	1005
Macro avg.	1.0	1.0	1.0	1005
Weighted avg.	1.0	1.0	1.0	1005

**Table 16 cancers-14-04834-t016:** SVM confusion matrix.

	Predicted Yes	Predicted No
Actual Yes	676	1
Actual No	0	328

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
