# Peer review of "Use of Personalized Biomarkers in Metastatic Colorectal Cancer and the Impact of AI"

_cancers, 2022, doi:10.3390/cancers14194834_

Round 1
Reviewer 1 Report
This paper provides a review of biomarkers in metastatic colorectal cancer and some discussion of AI.
Re biomarkers - Important markers are covered. There is some discussion of biomarkers in the adjuvant setting, eg., dMMR in early CRC, which should be deleted. The section on RNAs is very much out of proportion in terms of clinical relevance r.g., vs short paragraphs on RAS, BRAF and HER2. The sections on immunescore and epigenomics are also overly long given current relevance to clinical practice.
Re AI - This is not really discussion of AI as a topic in metastatic CRC, rather the authors presenting some of their own original data.
Author Response
Response about biomarkers :
We have deleted discussion about adjuvant settings in early CRC
We have extended the discussions with new relevant information about genomics biomarkers. Immunoscore and epigenomics represent in our opinion the emergent way for identifying new biomarkers in mCRC, with clinical relevance in the appropriate future.
Response about AI : We have completed with a review of AI in metastatic CRC

Reviewer 2 Report
This is a very comprehensive review paper. It is organized, highly detailed. It is one of the few times that I personally learned more about a topic, so I think it is very well done.
Only comment I would have is to add some more information/discussion about the clinical relevance/use of some of these items. There are papers in the literature for some of the less described topics that can be used as references. Examples:
Histone modification - PMID: 31055775
DNA methylation - PMID: 25580099
miRNA is covered very well, but mention of liquid biopsies in general would be appropriate for clinical relevance: examples - PMID: 35646657, PMID: 35264786, PMID: 35284120, PMID: 30128304, PMID: 29700206
Author Response
1.We have added more information about Histone modification and DNA methylation
2.We have added more information about liquid biopsy.
